# T Cell Immunity and the Quest for Protective Vaccines against *Staphylococcus aureus* Infection

**DOI:** 10.3390/microorganisms8121936

**Published:** 2020-12-06

**Authors:** Erin I. Armentrout, George Y. Liu, Gislâine A. Martins

**Affiliations:** 1Lung Institute, Cedars-Sinai Medical Center (CSMC), Los Angeles, CA 90048, USA; erin.armentrout@cshs.org; 2Division of Pulmonary and Critical Care Medicine, CSMC, Los Angeles, CA 90048, USA; 3Collaborative to Halt Antibiotic-Resistant Microbes, University of California, San Diego, La Jolla, CA 92161, USA; gyliu@health.ucsd.edu; 4Department of Pediatrics, University of California, San Diego, La Jolla, CA 92093, USA; 5F. Widjaja Foundation Inflammatory Bowel and Immunobiology Research Institute (IBIRI), CSMC, Los Angeles, CA 90048, USA; 6Department of Biomedical Sciences, Research Division of Immunology, CSMC, Los Angeles, CA 90048, USA; 7Department of Medicine, Division of Gastroenterology, CSMC, Los Angeles, CA 90048, USA

**Keywords:** *Staphylococcus aureus*, vaccine, antibodies, T cell-mediated immunity, tissue-resident memory T cells

## Abstract

*Staphylococcus aureus* is a wide-spread human pathogen, and one of the top causative agents of nosocomial infections. The prevalence of antibiotic-resistant *S. aureus* strains, which are associated with higher mortality and morbidity rates than antibiotic-susceptible strains, is increasing around the world. Vaccination would be an effective preventive measure against *S. aureus* infection, but to date, every vaccine developed has failed in clinical trials, despite inducing robust antibody responses. These results suggest that induction of humoral immunity does not suffice to confer protection against the infection. Evidence from studies in murine models and in patients with immune defects support a role of T cell-mediated immunity in protective responses against *S. aureus*. Here, we review the current understanding of the mechanisms underlying adaptive immunity to *S. aureus* infections and discuss these findings in light of the recent *S. aureus* vaccine trial failures. We make the case for the need to develop anti-*S. aureus* vaccines that can specifically elicit robust and durable protective memory T cell subsets.

## 1. Introduction

*Staphylococcus aureus* is a Gram-positive facultative anaerobe that is a leading cause of bacterial infections worldwide. It is especially problematic in hospital settings where *S. aureus* is the most common cause of post-operative infections [1] with costs that can exceed $90,000 per case [2]. In community settings, *S. aureus*, which includes both methicillin-sensitive (MSSA) and methicillin-resistant (MRSA) strains, is the most frequent cause of skin and soft tissue infection. The distinction between healthcare-associated (HA-MRSA) and community-associated *S. aureus* (CA-MRSA) infections is important because the prevalence of antibiotic resistance genes and certain virulence factors vary between the two types. This difference can affect infection severity and antibiotic management [3]. Overall, it is estimated that MRSA infections cause an annual burden of up to $13.8 billion in the US alone [4].

*S. aureus* has predilection for a broad range of tissues, which leads to manifestations such as folliculitis as well as life-threatening infections including pneumonia, endocarditis, osteomyelitis, and sepsis. The pathogen causes recurrent infections, suggesting that humans do not naturally develop long-lasting robust protective immunity against this bacterium. *S. aureus* also commonly colonizes human skin and mucosal surfaces, particularly in upper airways with some strains also showing a preference for the gastrointestinal tract [5]. It is estimated that about 30% of the population [6] is colonized with *S. aureus* and that colonization can occur as early as the neonatal period [7]. Colonization with *S. aureus* in healthy individuals is a risk factor for future staphylococcal infections by serving as the infectious source [8] and possibly by attenuating the host immune response against the pathogen.

*S. aureus* was first discovered in 1880 when Sir Alexander Ogston isolated the bacterium from an infected wound of a patient. The mortality rate for moderate to severe *S. aureus* infection was 80% in the pre-antibiotic era [9] and came down to 20% after the introduction of penicillin [10]. However, with each newly-introduced antibiotic, *S. aureus* promptly gained resistance and there is evidence that MRSA strains are associated with higher morbidity and mortality rates compared to their antibiotic susceptible counterparts [11].

The high disease burden and spread of antibiotic-resistant strains that led to antibiotic overuse has further highlighted the need for vaccination against *S. aureus*. Over the past decades and coinciding with the emergence of CA-MRSA, there has been a significant effort to develop anti-staphylococcal vaccines. Unexpectedly, to this date, every vaccine developed against *S. aureus* has failed in clinical trials, despite their demonstrated efficacy in animal models [12,13,14]. There is no consensus on why the vaccines have failed. Here, we review the immune response to *S. aureus* infection as it relates to vaccination and the failed vaccines, while making a case for alternative approaches to address this bottleneck to solving the staphylococcal health burden.

## 2. Previous and Ongoing Human *S. aureus* Vaccine Trials

There have been numerous attempts to create vaccines against *S. aureus*, which can be divided into two different approaches: Transfer of antibodies specific to staphylococcal antigens (passive immunization) or vaccination with recombinant antigens with the main goal of generating protective antibody responses (active immunization). These strategies either targeted surface proteins to promote the opsonophagocytic killing of *S. aureus* or targeted toxins to reduce cytolytic damages and immunopathology. *Staphylococcal* surface antigens targeted in vaccines include Clumping factor A and B (ClfA and ClfB), Iron-regulated surface determinant A and B (IsdA and IsdB), Capsular Polysaccharide 5 and 8 (CP5 and CP8), Serine–aspartate repeat protein D and E (SdrD and SdrE), and Manganese binding protein C (MntC). These proteins were selected on the basis of their high level of conservation across *S. aureus* strains [15]. Commonly selected *S. aureus* toxins included the Panton-Valentine Leukocidin (PVL, comprised of LukS-PV and Luk-PV), staphylococcal enterotoxin A and B (SEA and SEB), and α-toxin (Hla). Early vaccine studies focused on single antigens, whereas later studies targeted multiple staphylococcal antigens aiming to increase vaccine efficacy [15,16].

None of the vaccination or antibody transfer strategies clinically tested so far (recently reviewed by Redi et al. and Fowler & Proctor [13,14]) have shown protection or lasting protection against *S. aureus* infection in patients (Table 1 and Table 2) and the underlying reasons remain unclear. It has been suggested that the reasons underlying these failures range from the wrong choice of target antigens and adjuvants, to particularities of the study designs, target population inadequacy, and the notion that the murine experimental models of *S. aureus* infection used in pre-clinical vaccine studies likely do not fully reproduce the features of the human immune response to *S. aureus*. Regarding the latter, one important aspect to consider is the difference in colonization with *S. aureus* between mice and humans. In contrast to humans, which can be colonized with *S. aureus* early in life [7], mice are not colonized or are colonized at a lower frequency [17,18].

Nine different clinical trials testing passive immunization made it to phase II and III (Table 1). However, the great majority of these trials showed that antibody administration had no effect or offered only moderate protection in comparison to placebo treatment. Two studies utilized human polyclonal antibodies against CP5 and CP8, which was effective in shortening the length of stay for patients with bacteriemia but not in preventing invasive infection in neonates [19,20]. Another study tested if donor derived antibodies against ClfA from *S. aureus* and SdrG from *Staphylococcus epidermidis* could prevent sepsis in infants, but results showed no difference between placebo and experimental groups [21]. A single chain variable fragment targeting Adenosine triphosphate(ATP)-binding cassette (ABC) transporter GrfA was shown as ineffective in phase II clinical trials in another study [14]. The remaining five studies utilized monoclonal antibodies against *S. aureus* proteins. One study found that the administration of α-ClfA antibodies did not offer significant protection [22]. Similarly, monoclonal antibodies targeting Hla, PVL, LukED, and γ-hemolysin AB and CB (HlgAB, HlgCB) failed to protect against infection in mechanically-ventilated patients [23]. A monoclonal antibody targeting lipoteichoic acid (LTA) did not significantly protect against sepsis in low birthweight infants [24]. Another study tested a modified monoclonal antibody against Hla with an extended half-life to protect against *S. aureus* infection in mechanically ventilated patients. Unfortunately, treatment with this antibody also failed to confer protection to the treated group [25]. A more recent trial (Aridis Pharmaceuticals) is testing if the transfer of monoclonal antibodies against *S. aureus* Hla can treat community-acquired bacterial pneumonia or ventilator-associated pneumonia in patients, but this is still an ongoing trial and the results are not known. Overall, so far, the clinical trials concur with observations from murine model studies (see below), which indicate that antibody responses alone are not sufficient to protect against *S. aureus* re-infection. Other arms of the adaptive response, such as T cells, play important roles in protecting against this bacterium and therefore need to be engaged by vaccination approaches.

The notion that T cell immunity needs to be harnessed in vaccine development is also supported by observations from recent active immunization trials (Table 2), such as the failed V710 vaccine phase II/III clinical trial carried out by Merck in 2011. Vaccination with V710, an IsdB recombinant protein vaccine, generated abundant anti-IsdB antibodies however, the rate of *S. aureus* infection was comparable between placebo and vaccinated groups [27]. Additionally, antibody titers in patients who developed *S. aureus* infection were similar to those who did not develop the disease [28]. An analysis of cytokines associated with T cell function revealed that patients who developed fatal *S. aureus* infections had lower expression of IL-2 and IL-17A than those who survived the infection. In addition to the V710 trial, two other active staphylococcal vaccines that advanced to phase II/III clinical trials, StaphVax (Nabi) targeting CP5/CP8, and SA-4Ag (Pfizer, New York, NY, USA) targeting four staphylococcal antigens, also demonstrated a lack of efficacy in spite of robust antibody induction. Together, these observations support the importance of refocusing vaccine approaches to engage other arms of the immune system beyond the humoral response.

## 3. Innate Immune Responses to *S. aureus*

The innate immune responses to *S. aureus* and the pathogen’s strategies to counteract these responses have been the topic of many other recent and comprehensive reviews [36,37]. Here, we briefly summarize the innate response to *S. aureus* and focus mainly on the adaptive immune response as we discuss the new findings that can potentially inform the development of new vaccines. The innate immune system broadly recognizes invading pathogens through receptors that ligate pathogen-related molecular patterns (PAMPs), including single stranded RNA, lipopolysaccharide (LPS), and peptidoglycan (PGN). PAMPs are bound via pattern recognition receptors (PRRs), i.e., Toll-like receptors (TLRs), NOD-like receptors (NLRs), and/or C-type lectin receptors (CLRs). PRRs are mainly expressed by innate immune cells such as macrophages, monocytes, dendritic cells (DCs), and neutrophils. The sensing of PAMPs by TLRs initiates a signaling cascade that involves adaptor proteins myeloid differentiation primary response 88 (Myd88) or Toll/interleukin-1 receptor-domain-containing adapter-inducing interferon-β (TRIF). This is followed by interleukin-1 receptor-associated kinases (IRAKs), and nuclear factor kappa-light-chain-enhancer of activated B cells (NF-kB) or mitogen-activated protein kinase (MAPK) with a subsequent governance of pro-inflammatory gene expression. Alternatively, NF-kB activation by NOD2 signaling is modulated by receptor-interacting serine/threonine protein kinase 2 (RICK or RIP2).

The major *S. aureus*-associated PAMP sensed by the innate immune system appears to be LTA in addition to other lipoproteins, PGN, cell wall glycopolymers, RNA, and unmethylated CpG DNA. The recognition of *S. aureus*-LTA relies on the heterodimer, TLR2-TLR6, which senses diacylated lipoproteins such as LTA. Other receptors that recognize *S. aureus* PAMPs include NOD2, mannose-binding lectin, L-ficolin, surfactant protein A, TLR8, TLR9, and heterodimer TLR2-TLR1. The importance of TLR2, Myd88, IL-1 signaling, and NOD2 is underlined by several knockout mice studies. Knockout of TLR2, Myd88, IL-1R, or NOD2 increases the bacterial burden during infection compared to wild-type controls [38,39,40]. The importance of TLR, Myd88, and IL-1R signaling is mirrored in humans. Patients with defects in TLR, Myd88, IRAK4, and IL-1R signaling have higher incidence of *S. aureus* infections [41,42,43,44].

The binding of LTA by TLR2-TLR6 occurs on macrophages/monocytes as well as DCs. This triggers the expression/release of pro-inflammatory cytokines, such as IL-1b, IL-8, IL-10, IL-12, TNF-α and chemoattractants, MIP-1α, MCP-1, granulocyte colony-stimulating factor (GM-CSF), leukotriene B4, and complement factor 5a [45]. Chemoattractants recruit neutrophils to the site of infection, which clear the invading bacteria, while the pro-inflammatory cytokines act as chemoattractants as well as help shape the immune response [45] (Figure 1). After encountering *S. aureus*, DCs and macrophages travel from the site of infection to the draining lymph node where they can present *S. aureus* antigens to T cells. At least one study showed that that deletion of DCs by *S. aureus* α-toxin prevented the formation of a protective immune memory in a skin infection model [46], suggesting that DC-mediated antigen presentation is crucial for the development of successful adaptive immune responses against *S. aureus* (see below). The production of cytokines, as well as expression of co-stimulatory molecules by DCs help to initiate and direct the functional differentiation of T cell subsets that will ultimately shape the adaptive response to the infection.

In addition to DCs and macrophages, neutrophils have also been shown to play crucial roles in the immune response against *S. aureus*. Patients with neutropenia or defects in neutrophil function have higher incidences of *S. aureus* infections [47,48,49,50]. In animal models, it has been demonstrated that the recruitment of neutrophils leads to a decrease in bacterial burden, thus identifying neutrophils as a main player in clearing bacteria during infection. Neutrophils kill bacteria in several ways, such as via the creation of reactive oxygen species (ROS), production of antimicrobial peptides (AMPs), release of neutrophil extracellular traps (NETs), and degradation of bacteria through proteinases and hydrolases [51,52,53]. As we discuss below, although the innate immune response to *S. aureus* is important for controlling bacterial burden during infection, innate signaling upon natural infection appears inadequate to instruct the development of fully protective adaptive T and B cell responses against the bacteria. Work from our group has shown that manipulation of peptidoglycan O-acetylation leads to cell-wall resistance to lysozyme degradation thus shielding many staphylococcal cell-wall associated PAMPs from host detection. This results in the impaired production of various pro-inflammatory cytokines including IL-1b, IL-6, IL-23, and suboptimal development of protective Th1 and Th17 responses [54].

## 4. Adaptive Immune Response to *S. aureus*

### 4.1. B Cell Responses

Infection or colonization with *S. aureus* leads to the production of antibodies against both surface-bound proteins [55] and staphylococcal toxins [56,57]. These anti-staphylococcal antibodies are of various subclasses, i.e., IgM, IgA, and IgG. However, similarly to other successful pathogens, *S. aureus* has acquired strategies to evade host antibody responses. Two staphylococcal toxins, Surface Protein A (SpA) and second immunoglobulin binding protein (Sbi) can bind the Fc region of antibodies, which can prevent opsonization-mediated phagocytosis [58]. Additionally, both proteins interfere with complement activation to prevent opsonization [58]. SpA can act also as a B cell superantigen by binding the F(ab)2 portion of the B cell receptor, which leads to the inhibition of antibody production by inducing cell death [59].

A few studies have noted that antibody titers against *S. aureus* toxins correlate with better disease outcome. One study showed that the risk of sepsis was lower in individuals who had higher levels of IgG against α-hemolysin (Hla), δ-hemolysin (Hld), Panton Valentine leukocidin (PVL), staphylococcal enterotoxin C-1 (SEC-1), and phenol-soluble modulin α3 (PSM-α3) [60]. Another study looked at acute and convalescent antibodies levels in individuals with different types of infections, including first time skin and soft-tissue (SSTI), recurrent SSTI, and invasive infections, such as bacteremia, osteomyelitis, septic arthritis, bursitis, pyomyositis, empyema, endocarditis, or septic thrombophlebitis [61]. The patient group that had the highest convalescent titers was the invasive infection cohort. This group had an increase in production of α-Hla antibody titers, which correlated with protection against subsequent infection within 12 months. In contrast, SSTI did not induce long-lasting antibody production. These observations suggest that the route of infection can affect the type of humoral immune response mounted against *S. aureus*. Additionally, our recent study showed that the adoptive transfer of sera from children with invasive staphylococcal diseases protected naïve mice from systemic staphylococcal infection [62]. However, protection conferred by the sera was only observed with some of the convalescent serum samples and sera collected at 6 months post-infection were not protective, albeit the size of the study was small. These antibodies most likely neutralized *S. aureus* toxins, which in addition to preventing damage caused by the toxin also prevented progression of infection. Another study in mice found that serum transfer after vaccination against ClfA, ClfB, Hla, IsdA, IsdB, and SdrD prevented bacteremia and reduced dermonecrosis but did not reduce the bacterial burden in abscesses or GI colonization [63].

Although the studies discussed above indicate that B cells are capable of eliciting anti-staphylococcal responses, one of our most recent studies showed that antibodies were dispensable for protective immunity against *S. aureus* in mice vaccinated with four different *S. aureus* recombinant proteins (ClfA, IsdA, MntC, and SdrE) encapsulated in β-glucan particles [64]. Importantly, B cell-deficient mice and patients with X-linked agammaglobulinemia, a genetic disorder that results in severe B cell deficiency, do not have increased susceptibility to *S. aureus* infection [65,66]. Despite these observations, the majority of vaccine studies, including several clinical trials (Table 2) have focused on antibody production as a measure of induction of protective immunity. As we discuss below, there is strong evidence that T cell-mediated immunity is required for long-lasting protection against *S. aureus* infection. Thus, the data so far support the notion that the engagement of certain T helper cell subsets, which are usually suppressed by natural infection, is a key component for the development of protective immunity to re-infection, and therefore should be the basis of novel approaches for vaccine development.

### 4.2. T Cell Responses

T cells make up the other branch of the adaptive immune response and are defined by the expression of the T cell receptor (TCR) which can either be comprised of gamma and delta chains (γδTCR) or alpha and beta chains (αβTCR). One difference between these cell subsets is that γδ-T cells do not have a broad clonal diversity as αβ-T cells, and thus epitope coverage by αβ-T cells is wider compared to γδ-T cells.

#### 4.2.1. Roles of γδ T Cells

Several studies have indicated that γδ-T cells play important roles in the control of *S. aureus* infection. One study described more severe skin pathology in γδ-T cell knockout mice, which was attributed to decreased IL-17 production [67]. Another study reported the expansion of certain γδ-T cell clones after infection. Furthermore, the adoptive transfer of γδ-T cells from mice previously infected with *S. aureus* protected naïve mice from infection, suggesting that γδ-T cells could provide some sort of protective memory to *S. aureus* infection [68]. However, in that study, the γδ-T cells were primed in the absence of IL-1β, which differs from what happens during natural *S. aureus* infection. Moreover, as measured by lesion size and bacterial burden, protection induced by the adoptively transferred γδ-T cells was modest and only apparent during early time points after infection. Nonetheless, there is precedence for a role of “memory” γδ-T cells in other bacterial infection models. For example, “memory” Vg4^+^ γδ-T cells have been shown to provide IL-17-dependent protection against *Listeria monocytogenes* oral infection [69]. Thus, it is conceivable that γδ-T cells could contribute to memory responses to *S. aureus* infection in specific circumstances, but this remains to be validated.

#### 4.2.2. Roles of αβ T Cells

αβ-T cells make up the majority of T cells and their functions can be defined on the basis of their expression of the surface molecules CD4 and CD8. CD4^+^ T cells, also called helper T (Th) cells, contribute to immune responses against infections by providing help to B cells and CD8^+^ T cells. In contrast, CD8^+^ T cells, also called cytotoxic T cells, lyse cells that are infected by pathogens. CD4^+^ T cells have critical roles in the defense against extracellular pathogens, whereas CD8^+^ T cells preferentially recognize and kill intracellular/viral pathogens and tumor cells. CD4^+^ helper T cells can differentiate into different subsets, including Th1, Th2, and Th17, among others [70]. These Th subsets are mainly defined by the cytokines they produce and their role in certain immune responses. Th1 cells depend on the transcription factor, T-bet, for differentiation. They preferentially produce IL-2 and IFNγ and have been shown to help protect against intracellular pathogens [70]. The development of Th2 cells requires the transcription factor GATA-3, which is important to mediate the expression of the majority of Th2-associated cytokines, including IL-4, IL-5, and IL-13 [70]. Th2 cells are associated with humoral immunity and parasitic infections. Th17 cells require the expression of RORγt and several other transcription factors [70] and produce primarily IL-17 family cytokines (IL-17A, F, C, and E) in addition to IL-22 and, in some cases, GM-CSF. Th17 cells have been shown to mediate protection against extracellular pathogens such as bacteria and fungi [70]. In addition, Th cells can also originate T follicular helper cells (T_FH_), which require the transcription factor BCL-6. T_FH_ produce cytokines, such as IL-21, which promote B cell responses [70]. The differentiation of Th cells can also result in the generation of Foxp3^+^ regulatory T cells (termed inducible T_REG_ cells), which have similar functions as thymic derived (t) T_REG_ [70].

Evidence to date suggests that both CD4^+^ and CD8^+^ T cells are important for immunity against *S. aureus* infection. Investigations have focused on CD4^+^ T cells because *S. aureus* is traditionally considered an extracellular pathogen. However, there is data indicating that once internalized by cells such as macrophages, *S. aureus* escapes phagosomes to enter the cytoplasm, and therefore becomes an intracellular pathogen [71] that requires control by CD8^+^ T cells or NK cells. In contrast, studies in murine models indicate that CD8^+^ αβ-T cells are dispensable during *S. aureus* infection [72,73], whereas there is an abundance of data, in empyema and skin infection models, supporting a central role of CD4^+^ T cells. Of note, *S. aureus* has developed several virulence factors to counteract CD4^+^ αβ-T cell responses including the superantigens staphylococcal enterotoxin A (SEA), enterotoxin B (SEB), and toxic shock syndrome toxin (TSST-1), which bind major histocompatibility complexes (MHC) II and T cell receptors (TCR). This binding of MHC class II and TCR activates the uncontrolled release of inflammatory cytokines from T cells, followed by the induction of T cell anergy or deletion. Additional staphylococcal strategies consist of the expression of an MHC class II analog protein (Map), which binds directly to TCR to attenuate T cell responses [74], and the expression of the LukED toxin that lyses T cells following ligation to the CCR5 receptor [75]. The importance of CD4^+^ T cells in the immune response against *S. aureus* in humans is supported by the observation that HIV patients with decreased CD4^+^ T cells have increased prevalence of *S. aureus* infections [76,77,78].

Studies of the mechanisms underlying CD4^+^ T cells roles in *S. aureus* infection indicate that IFNγ may be dispensable during primary infection, whereas the requirement for IL-17 varies depending on the infection model. One study found that IFNγ knockout mice had higher survival rates and lower bacterial burdens during intravenous infection compared to wild type controls [79]. Similarly, another study found that there was no effect on primary skin *S. aureus* infection in IFNγ-receptor knockout mice, suggesting that IFNγ signaling is not required for bacterial clearance in the skin [67]. In contrast, primary cutaneous infection in IL-17-receptor knockout mice resulted in larger lesions and bigger bacterial burdens compared to wild type mice, whereas IL-17-deficient mice had mortality rates comparable to that observed in wild type mice during intravenous infection [80,81]. Thus, during primary *S. aureus* infection, IFNγ seems to be dispensable while IL-17 is only conditionally required.

In contrast to the observed in primary infection, it is well established that both Th1 and Th17 CD4^+^ T cells are required to mediate protection against *S. aureus* in different models of re-infection in mice. One study found that the transfer of IFNγ producing αβ-CD4^+^ T cells protected against *S. aureus* peritoneal infection by a mechanism that involved the activation of peritoneal macrophages [82]. Likewise, protection against *S. aureus* peritoneal infection was conferred by an IL-17-dependent mechanism in another vaccination study [83]. Lin et al. also showed that vaccination with recombinant N-terminus of the *Candida albicans* adhesin protein Als3p, which is structurally similar to *S. aureus* ClfA, induced IFNγ and IL-17A single- and double-producing CD4^+^ T cells that protected against *S. aureus* bloodstream infection. In this model, protection was associated with increased neutrophil influx, which reduced bacterial burden [84].

In a different study, vaccination with a multiple antigen presenting system with six staphylococcal proteins led to the differentiation of CD4^+^ T cells that produced both IFNγ and IL-17A, which mediated protection against both bloodstream and skin infections [63]. Additionally, in a study showing that immunization intravenously could protect mice against subcutaneous infection, this protection was mediated by CD4^+^ T cells and was associated with an increase in IFNγ production [46]. In one of our recent studies, we found that the vaccination of mice with β-glucan particles loaded with four different proteins from *S. aureus* also generated an innate immune response that fostered the development of Th1 and Th17 cells that could transfer protection to naïve mice [64].

Although the requirement for IL-17 in the immune response to *S. aureus* is clearly illustrated by the studies mentioned above, the cellular source of this cytokine during *S. aureus* infection is less clear with conflicting reports showing a role for both αβ and γδ T cells. Cho et al. suggested that IL-17 was produced by γδ-T cells, which recruited neutrophils for the control of *S. aureus* primary skin infection [67]. In contrast, Montgomery et al. found that IL-17A and CD4^+^ T cells were the key to controlling *S. aureus* infection in the skin [73]. Additionally, Marchitto et al. showed that upon skin infection, both CD4^+^ and γδ-T cells produced IL-17A/F at the site of infection as well as in the draining lymph node [68]. However, they concluded that the expression of IL-17A was higher in γδ-T cells. Thus, both αβ-CD4^+^ T cells as well as γδ-T cells can produce IL-17 and be required to control *S. aureus* infection. The importance of IFNγ and IL-17 in the response to *S. aureus* infection in humans is supported by several studies [79,80,81,85]. The role of IL-17 is further indicated by the finding that individuals with genetic defects that interfere with the differentiation of Th17 cells, such as STAT3 loss-of-function mutations or hyper-IgE syndrome, have a higher incidence of *S. aureus* infection [86,87,88].

In addition to *S. aureus*, Th17 cells and IL-17 are crucial in the response to several other extracellular bacterial infections. Likewise, Th1 cells and IFNγ play important roles in the response to many intracellular pathogens. Although advantageous in the context of these infections, both IL-17 and IFNγ are highly inflammatory cytokines and can cause unwanted harm to host tissues. Consequently, Th1 and Th17 responses must be tightly regulated. The immune system has evolved to deploy several different mechanisms to control Th1 and Th17 responses. One important mechanism used to control IL-17 and IFNγ actions is the expression of the regulatory cytokines TGFβ and IL-10, which can be produced by many different immune cell types. The role of TGFβ in the immune response to *S. aureus* has not been extensively evaluated, but at least one study showed that culture of bovine mammary epithelial cells with heat-killed *S. aureus* leads to the expression of TGFβ [89]. Another study showed that partial genetic deletion of TGFβ Receptor 2 (Tgfbr2) or the administration of a TGFβ receptor inhibitor in mice was associated with improved bacterial clearance in the skin [90]. These data suggest that *S. aureus*-induced TGFβ could play a role in damping the responses against the infection in the skin, but this remains to be further studied.

The role of IL-10 in the immune response to *S. aureus* infection is not fully understood. In studies looking at systemic infection, IL-10 appears to be protective; IL-10 deficient mice had higher bacterial burdens compared to wild type controls [91,92]. Additionally, one study found that the incidence of *S. aureus*-induced arthritis was higher in IL-10-deficient mice [92]. In contrast, during skin infection, IL-10-deficient mice had smaller lesion sizes and lower bacterial burdens compared to wild type counterparts [91]. These data suggest the complex role of IL-10 during *S. aureus* infection may depend on the route of infection.

Importantly, when naïve human T cells were cultured with *S. aureus*, they produced IL-10 in addition to IL-17 [93]. However, if the T cells were cultured with *S. aureus* along with IL-1β, the cells secreted IL-17 and IFNγ, but not IL-10. This was in stark contrast to what was observed with *C. albicans*, which caused human T cells to produce IL-17 and IFNγ and very little IL-10 without any manipulation to the culture conditions. An additional study in human samples observed that higher IL-10 serum levels correlate with higher rates of mortality in patients with bacteremia caused by *S. aureus* [94], suggesting that it can play an inhibitory role in human immunity. Together, these observations suggest that *S. aureus* can somehow manipulate the immune response to prevent the differentiation of IL-17 and potentially IFNγ-producing cells and/or undermine the actions of these cells. In line with that, our recent study showed that O-acetylation of peptidoglycan by *S. aureus* leads to a preferential expression of IL-10 instead of IL-17 or IFNγ. Furthermore, deletion of O-acetyl transferase in *S. aureus* increased the expression of cytokines associated with Th17 polarization and contributes to protection from intraperitoneal reinfection via an IL-17-dependent mechanism [54]. We also found that immunized IL-10-deficient mice had improved bacterial clearance upon intraperitoneal infection with *S. aureus*, and importantly, protection induced by genetic deficiency of IL-10 could be transferred to naïve mice by the adoptive transfer of CD4^+^ T cells. Thus, the expression of IL-10 by CD4^+^ T cells upon primary infection with *S. aureus* is likely one of the factors contributing to the lack of robust long-lasting protective CD4^+^ T cell memory responses.

The cellular source of IL-10 and the exact mechanisms by which IL-10 shapes effector and potentially memory CD4^+^ T cells responses to *S. aureus* infection remain to be elucidated. Considering the evidence above, it is tempting to speculate that the use of adjuvants that could counteract the induction of IL-10 by *S. aureus* could be a promising approach for the development of new vaccines to favor the induction of robust Th17 and Th1 responses and potentially generate long-lasting protective immunity against *S. aureus*.

#### 4.2.3. Tissue Resident Memory T Cells

Tissue-resident memory T cells (T_RM_) are a recently described memory T cell subset that is thought to reside in tissues long-term. Similar to other T cell memory subsets, T_RM_ respond rapidly and specifically upon re-infection [95], but have the added advantage to be present at the site of injury and therefore could provide a fast and efficient response against insults. The exact mechanisms underlying the differentiation of T_RM_ are not fully understood. Studies using different infection models [95,96,97], indicate that T_RM_ cells develop from memory precursors that enter the tissue during the effector phase of infection and remain in the tissue afterwards [98]. However, the exact regulation of this process is unknown.

One mechanism by which tissue retention of T_RM_ is thought to occur is through the antagonism of sphingosine 1-phosphate receptor-1 (S1PR1) by the surface molecule CD69, also called early T cell-activation antigen P60 [99]. S1PR1 regulates the egress of T cells from tissues and CD69 likely binds S1PR1 directly, causing a conformational change that interferes with its function [100]. Of note, the expression of S1PR1 along with other genes involved in secondary lymphoid organ (SLO) recirculation are regulated by the transcription factor Krupple-like factor 2 (KLF2), which is downregulated in T_RM_ cells [101]. Additionally, if the expression of S1PR1 is induced outside of its typical regulation, it reduces the frequency of cells localizing to non-lymphoid tissues. CD103 (integrin αE) and CD49a (integrin α1) are other surface markers associated with the T_RM_ phenotype that might aid in the retention of these cells in the tissues. CD103 binds E-cadherin with integrin beta 7 and is thought to enable the cells to enter and remain in the tissue. CD49a binds collagen along with CD29 (integrin beta 1) to help retain cells in collagen-rich tissues [102]. Additionally, T_RM_ lack the expression of the chemokine receptors CCR7 and CD62L, which are required for homing to lymph nodes.

Several studies have detailed the importance of T_RM_ during re-infection at specific sites such as, the lungs, skin, and female reproductive tract. Our current knowledge of T_RM_ biology is largely based on studies focusing on CD8^+^ T_RM_ cells residing in the tissue long-term, with many fewer studies detailing the characteristics of CD4^+^ T_RM_. Therefore, the bulk of the data discussed below is based on CD8^+^ studies. Similarities and differences with CD4^+^ T_RM_ are highlighted/discussed when appropriate.

As alluded to above, the mechanisms underlying T_RM_ cell differentiation and function are poorly defined. In addition to KLF2, mentioned above, there are several other transcription factors that may play a role in the establishment of T_RM_ tissue residency, including Blimp-1, Hobit, T-bet, Eomes, and Runx3. Mackay et al. found that when the transcription factors Hobit and Blimp-1 were knocked out, significantly fewer T_RM_ cells were found in the skin, gut, liver, and kidney in herpes simplex virus (HSV) and lymphocytic choriomeningitis virus (LCMV) experimental infection models [103]. This study indicated that Blimp-1 and Hobit work together to modify transcriptional profiles that are required for T_RM_ differentiation and potentially, for function. Similarly, deletion of the transcription factor Runx3 prevented the formation of T_RM_ cells, especially CD69^+^ CD103^+^ T_RM_ cells localized to the gut after LCMV infection [104]. This study concluded that Runx3 was a transcriptional regulator that upregulated core genes responsible for tissue residency and downregulated genes important for circulating properties. Additionally, the downregulation of two other transcription factors, T-bet and Eomes, caused by TGF-β signaling, was seen in T_RM_ cells migrating to the skin after HSV infection [105]. However, a complete ablation of T-bet resulted in the loss of these cells over time. This was likely because of the low expression of the IL-15 Receptor (IL15R) β chain in these cells. IL-15 signaling is necessary for the long-term survival of T_RM_ cells in the skin.

How cytokine signaling determines the fate of tissue-resident memory T cells is not well understood however, several studies have suggested that TGFβ plays a role in programming T cells to home to and reside in peripheral tissues. TGFβ stimulation of activated CD8^+^ T cells in vitro causes a gene regulation signature that closely resembles that seen in T_RM_, which suggests that the T_RM_ genetic signature could be regulated in part by TGFβ signaling [106]. One study demonstrated that TGFβ released from DCs in the lymph nodes before infection predisposed CD8^+^ T cells to home to the tissues after antigen stimulation [107]. Similarly, Thompson et al. found that TGFβ released by monocytes caused the upregulation of CD103 on naïve T cells, which helped the T cells to migrate to peripheral tissues, such as the lung [108]. Additionally, TGFβ was required for maintaining T_RM_ within the intestines, which was linked to the induction of CD69 [109].

Although the vast majority of T_RM_ studies in murine models focus on CD8^+^ T cells, at least three different studies indicate that CD4^+^ T_RM_ cells are important for protection against bacterial infection. Wilk et al. showed that CD4^+^ T_RM_ (CD44^hi^ CD62L^lo^ CD69^+^ CD103^+/−^) cells could be found in the lungs of mice after *Bordetella pertussis* infection and that these cells conferred protection against re-infection [110]. Another group demonstrated that CD4^+^ T_RM_ found in the lungs can protect against *Klebsiella pneumoniae* infection without the help of circulating CD4^+^ T cells [111]. Importantly, a 2015 study by Stary et al. found that vaccination-induced protection against *Chlamydia trachomatis* infection was mediated at least in part by CD4^+^ T_RM_ cells induced in the female reproductive tract of mice [112]. This result suggests that vaccines can be designed to manipulate the immune response to optimize the induction of T_RM_ cells that can protect against bacterial infection.

T_RM_ surface markers identified in humans correspond with surface markers found in murine T_RM_, including CD69, CD103, and CD49a [113]. However, a direct examination of the function of these cells in patients has not been performed and thus, the requirement of T_RM_ for protection against bacterial infection in humans remains to be established. Nonetheless, at least a few studies indicate that in humans, both CD8^+^ and CD4^+^ T_RM_ are required for vaccine-induced protection against viruses. Studies performed in human tissues, including the lung and skin, identified the presence of CD4^+^ and CD8^+^ T cells with specificity to ubiquitous human pathogens such as influenza [114,115], herpes simplex virus (HSV) [116], cytomegalovirus (CMV) [117], and respiratory syncytial virus (RSV) [118]. Additionally, one study looking at RSV-specific T_RM_ cells correlated an increase in T_RM_ cells with lower viral titers [118], suggesting that T_RM_ cells help mediate the immune response upon re-infection, thereby providing protection.

The induction and functional requirement of T_RM_ for long-term protective immunity against *S. aureus* remains to be investigated. However, if the generation of T_RM_ cells upon primary infection with other bacteria is a requirement for protective immunity, as suggested by the data discussed above, then one could speculate that primary responses to *S. aureus* infection might not result in T_RM_ cell generation (Figure 2). The generation of skin-specific T_RM_ cells has been observed in several other infectious models, including *Leishmania major* [119], vaccinia virus (VACV) [120], LCMV [121], and HSV [122]. In these studies, T_RM_ were shown to provide protection against re-infection. Furthermore, Linehan et al. found that *S. epidermidis*-specific CD8^+^ T_RM_ cells were created after an application of a strain from a particular clade (NIHLM087 from the A20 clade) [123]. This study is of particular interest because *S. epidermidis* is closely related to *S. aureus* however, *S. epidermidis* is exclusively part of the microbiome, whereas *S. aureus* can cause pathogenic infections. Therefore, it is unlikely that the same vaccination strategy would work to induce the development of anti-*S. aureus* specific protective T_RM_ cells, but that remains to be tested. Nonetheless, the fact that protective T_RM_ responses can be generated against *S. epidermidis* is encouraging and might help to inform the criteria for the development of novel vaccination approaches against *S. aureus*. Future studies should focus on what aids in creating *S. aureus*-specific T_RM_ cells and the possible hurdles that need to be overcome to create *S. aureus*-specific T_RM_ cells.

## 5. Concluding Remarks and Future Directions

Despite promising pre-clinical studies, every anti-staphylococcal vaccine developed to date has failed in clinical trials. It is still unclear why these experimental vaccines have not translated into successful vaccines and what a successful vaccine would look like, but there have been notable advances in our understanding of how long-lasting protective immunity to pathogens develop. Moving forward, the field will benefit from this new understanding as well as of a more comprehensive characterization of T cell-mediated immunity to *S. aureus*, especially after secondary infections. This knowledge should provide a framework for the development of strategies to mitigate the manipulation of adaptive immunity by *S. aureus*, and thus inform the development of novel and efficacious vaccines. Clarifying if natural infection with *S. aureus* leads to the differentiation of antigen-specific T_RM_ and their potential function (or disfunction) in both experimental models and in patients is just one of the many aspects that will contribute to the development of a protective vaccine against *S. aureus*. The development of new tools that facilitate the characterization of the *S. aureus*-specific T cell responses, such as *S. aureus*-specific TCR transgenic mice, will also greatly benefit the field.

## Figures and Tables

**Figure 1 microorganisms-08-01936-f001:**
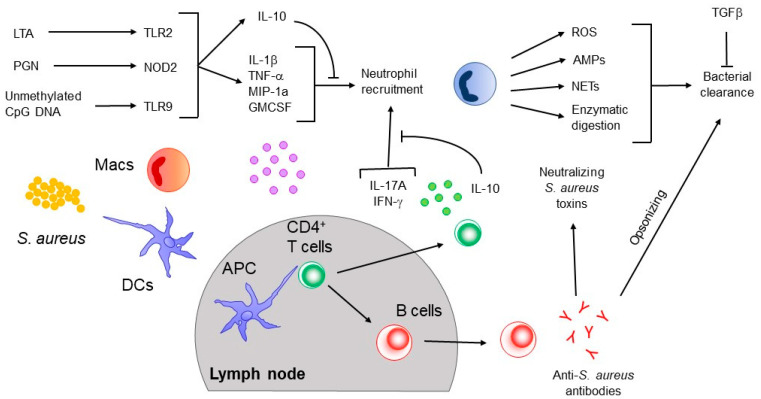
The immune response against *S. aureus*. Pattern recognition receptors (TLR2/NOD2/TLR9) expressed on innate immune cells such as macrophages (Mac) and dendritic cells (DC) at the infection site respond to *S. aureus* pathogen-related molecular patterns (LTA, peptidoglycan (PGN), and unmethylated CpG DNA). This interaction leads to the release of cytokines and chemokines (IL-1β, TNF-α, MIP-1α, and GM-CSF), which attracts neutrophils to the site of infection. Neutrophils mediate bacteria killing and clearance via reactive oxygen species (ROS), anti-microbial peptides (AMPs), neutrophil extracellular traps (NETs), and enzymatic digestion by hydrolases and proteinases. After interacting with *S. aureus,* Mac and DC can migrate to draining lymph nodes to present the antigen to naïve T cells. This interaction causes T cells to undergo functional differentiation, which culminates with the generation of IFNγ and IL-17-producing cells. IFNγ can activate Mac and DC to enhance antigen presentation (among other inflammatory functions). The expression of IL-17 facilitates neutrophil recruitment. The anti-inflammatory cytokines TGFβ and IL-10 can also be induced upon infection and could play both protective (limiting damage) or inhibitory (limiting immune response) roles, depending on the route of infection. Additionally, T_FH_ cells could differentiate in response to the initial infection and could interact with B cells so as to foster the production of anti-staphylococcal antibodies that enable the opsonization of bacteria or neutralize staphylococcal toxins.

**Figure 2 microorganisms-08-01936-f002:**
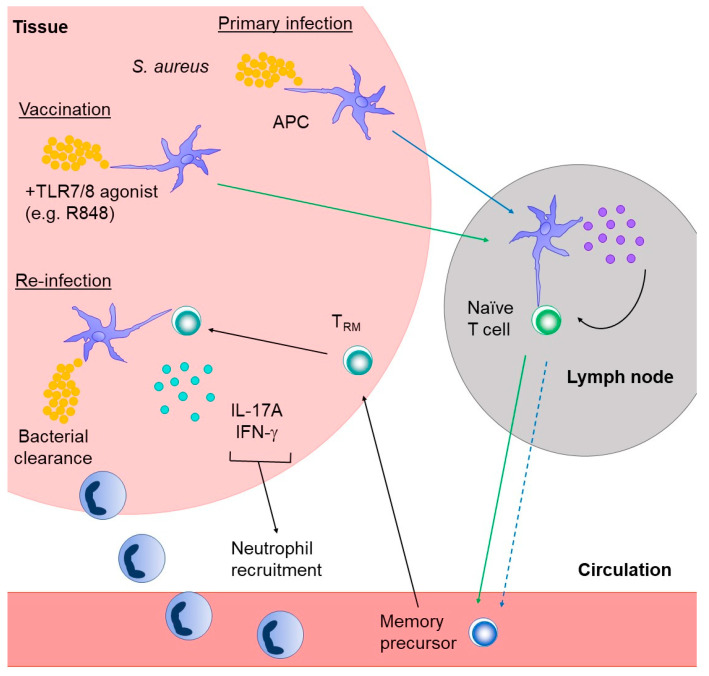
A model for potential roles of T_RM_ in protecting against *S. aureus* infection. Given the poor induction of long-lasting protective immunity to *S. aureus* infections, it is possible that natural *S. aureus* infection (broken blue arrows) fails to lead to the development of T_RM_ cells. Vaccination strategies (green arrows) proven to induce protective T_RM_ responses against another bacterial infection [112] could be employed to promote the induction of protective T_RM_ cells. If generated upon vaccination, *S. aureus*-specific T_RM_ cells could produce IL-17A and IFNγ leading to neutrophil recruitment and subsequent bacterial clearance upon re-infection of sites such as the skin and soft tissues.

**Table 1 microorganisms-08-01936-t001:** Clinical trials of passive *S. aureus* immunization.

Antigen	Company/ Name	Phase	Target Group	Result	Study Dates	Notes	Citation
CP5/CP8	Nabi/ Altastaph	II	Treatment of Bacteriemia	Completed	2002–2004	Shorter length of stay for treatment group. No further development.	Rupp et al., 2007 [20]
CP5/CP8	Nabi/ Altastaph	II	Prevention of invasive infection in very low birth weight neonates	Failed	2003–2004	No difference between placebo and treatment groups.	Benjamin et al., 2006 [19]
ClfA	Bristol-Myers Squibb/ Aurexis Tefibazumab	II	Treatment of bacteremia	Failed	2005	No differences between groups.	Weems et al., 2006 [22]
ATP-binding cassette (ABC) transporter GrfA	Neu Tec Pharma/Aurograb	II	Treatment of deep-seated infection	Failed	2004–2006	No difference between groups	Fowler and Proctor 2015 [14]
Hla	AstraZeneca/ Suvratoxumab/ MEDI4893	II	Mechanically ventilated adults	Failed	2014–2018	No significant differences between groups.	Yu et al., 2017 [25]
Hla, PVL, LukED, g-hemolysin AB and CB (HlgAB, HlgCB)	Arsanis Inc/ ASN100	II	Mechanically ventilated subjects	Failed	2016–2018	Did not achieve endpoint. Ended in futility.	Magyarics et al., 2019 [23]
ClfA (*S. aureus*) and SdrG (*S. epidermidis*)	Bristol–Myers Squibb/ Veronate/ Inhibitex	III	Sepsis in premature infants	Failed	2004–2006	No difference between placebo and treated group.	DeJonge et al., 2007 [21]
LTA	Biosynexus incorporate/ Pagibmaximab	II	Prevention of Sepsis in very low birth weight neonates	Failed	2009–2011	No significant difference between groups.	Weisman et al., 2011 [24]
Hla	Aridis Pharmaceuticals/ AR-301	III	Treatment of bacterial pneumonia or ventilator-associated pneumonia	Completed	2019–2020	Promising phase II trials (Francois et al., 2018) [26]	unpublished

**Table 2 microorganisms-08-01936-t002:** Clinical trials of active *S. aureus* immunization.

Antigen	Company/ Name	Phase	Target Group	Result	Study Dates	Notes	Citation
Enterotoxin A and C1, TSST, alpha-hemolysin, LukS, LukF	BioTherapeutics/ IBT-V02	Pre-clinical		No data	NA	Scheduled to enter phase I clinical trials in 2020	Aman 2018 [29]
rAT/rLukS-PV	Nabi/ N.A.	I		Completed	2009–2011	No further data	Landrum et al., 2017 [30]
Recombinant Enterotoxin B (rSEB)	BioTherapeutics/ STEBVax	I		Completed	2011–2015	No further data	Chen et al., 2016 [31]
Unspecified Recombinant protein—bioconjugated—adjuvanted	GSK/ GSK3878858A	I		Ongoing	2020–2020	No data	Unpublished
Als3p	NovaDigm Therapeutics/ NDV-3A	II	Nasal colonization (and SSTI) in military personnel	Completed	2018–2019	Well tolerated in phase I trials and provided protection against *S. aureus* and *C. albicans*	Schmidt et al., 2012 [32]
CflA/MntC/CP5/CP8	Pfizer/ SA-4Ag	IIb/III	Elective orthopedic surgery	Failed	2015–2019	Stopped after sub-par results	Gurtman et al., 2019 [33]
CP5/CP8	Nabi/ StaphVax	III	Renal disease or orthopedic surgery	Failed	2005–2006	No difference between placebo and vaccinated groups	Fattom et al., 2004; Fattom et al., 2015 [34,35]
IsdB	Merck/ V710	III	Prevention of infection after cardiothoracic surgery	Failed	2007–2011	Adverse outcomes in vaccinated group	Fowler et al., 2013; McNeely et al., 2014 [26,27]

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
