# Peer review of "T Cell Immunity and the Quest for Protective Vaccines against Staphylococcus aureus Infection"

_microorganisms, 2020, doi:10.3390/microorganisms8121936_

Round 1

Reviewer 1 Report

The manuscript presented by Armentrout and collaborators presents different aspects related to the generation of innate and adaptive immune responses in the context of infection caused by S. aureus. The authors describe very very well the mechanisms of immunity and the limitations related to the development of anti-S. aureus vaccines. I consider the manuscript to be very well written and important for readers in microbiology and development of vaccines against S. aureus.

Author Response

We thank the reviewer for their feedback.

Reviewer 2 Report

1.The title name is vague. It should be changed so that the reader can understand the association with the memory T cells claimed by the authors.

2.The names of microorganisms in this manuscript should be unified in italics.

3.Other than the above two points, there seems to be no particular problem.

Author Response

We thank the reviewer for their feedback. We have addressed their concerns in the revised version of the manuscript.

"1.The title name is vague. It should be changed so that the reader can understand the association with the memory T cells claimed by the authors."

We have changed the review title from “Developing a Protective Staphylococcus aureus Vaccine” to “T cell immunity and the quest for Protective Vaccines against Staphylococcus aureus Infection”

"2.The names of microorganisms in this manuscript should be unified in italics."

This has been corrected throughout the manuscript

"3.Other than the above two points, there seems to be no particular problem."

Reviewer 3 Report

Staphylococcus aureus is an opportunistic pathogen, which can develop antibiotic resistance and result in severe disease. Effective vaccination provides an alternative to antibiotics, ensuring protection against both antibiotic-resistant and sensitive infections. This paper discusses the results of the recent S.aureus vaccine trials and describes new approaches to develop anti-S.aureus vaccines. It was demonstrated that S.aureus vaccines that target humoral immunity alone do not provide sufficient protection from disease associated with this pathogen. However, activation of alternative T cells could provide a novel strategy for improving therapeutic efficacy of vaccine and development of long-lasting protective immunity against S.aureus. The review performed is systematic and thorough, and the conclusions reached by the authors are warranted by the experimental data discussed.

Author Response

We thank the reviewer for their feedback. We have addressed their concerns in the revised version of the manuscript. We have revised the manuscript to edit English language and style.